# GENERALIZED GREEDY GRADIENT-BASED HYPERPARAMETER OPTIMIZATION

## ABSTRACT

Bilevel Optimization (BLO) is a widely-used approach that has numerous applications, including hyperparameter optimization, meta-learning. However, existing gradient-based method suffer from the following issues. Reverse-mode differentiation suffers from high memory requirements, while the methods based on the implicit function theorem require the convergence of the inner optimization. Approximations that consider a truncated inner optimization trajectory suffer from a short horizon bias. In this paper, we propose a novel approximation for hypergradient computation that sidesteps these difficulties. Specifically, we accumulate the short-horizon approximations from each step of the inner optimization trajectory. Additionally, we demonstrate that under certain conditions, the proposed hypergradient is a sufficient descent direction. Experimental results on a few-shot meta-learning and data hyper-cleaning tasks support our findings.

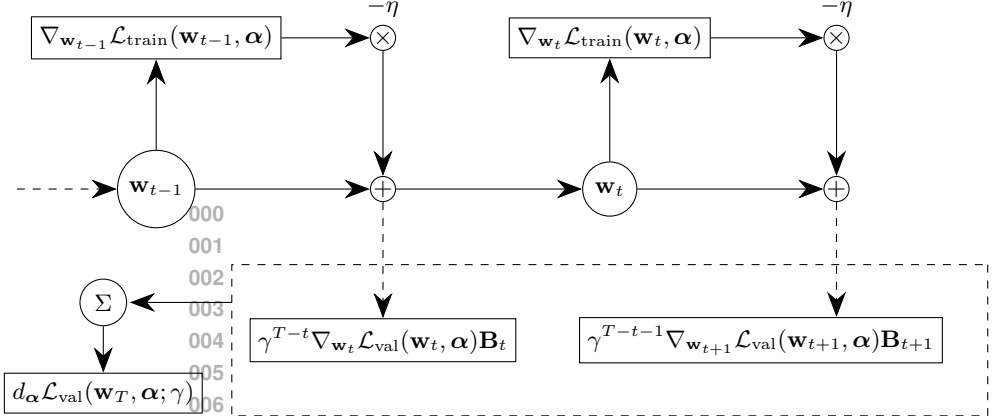

Figure 1: The schematic illustration of the proposed approach. In general, the approximate hypergradient is calculated as a weighted sum of the locally optimal greedy gradients calculated at each inner optimization step.

## 1 INTRODUCTION

Bilevel optimization has become an essential component of machine learning, which includes Neural Architecture Search Liu et al. (2018); Pham et al. (2018); Zoph & Le (2016), Hyperparameter Optimization Hutter et al. (2019), and Meta-Learning Hospedales et al. (2021); Nichol et al. (2018); Finn et al. (2017). In the hierarchical optimization framework, the outer-level objective is aimed to be minimize given the optimality in the inner level. Solving the bilevel problem is challenging due to the intricate dependency of the optimal inner parameters given the outer parameters.

Naive approaches such as random search and grid search Bergstra & Bengio (2012) become impractical with the growing number of hyperparameters to be optimized due to the curse of dimensionality. Another approach that has proven effective in low-dimensional setting is Bayesian Optimization

Snoek et al. (2012). However, its extension to high-dimensional setting is challenging Wang et al. (2023).

In the current work we develop a novel gradient-based algorithm Bengio (2000). The challenge is that the exact hypergradient calculation is computationally demanding Franceschi et al. (2017). Specifically, Forward-Mode differentiation (FMD) is memory demanding, since it increases linearly with the number of hyperparameters. This limits the application of the method for large-scale problems with millions of hyperparameters, such as meta-learning. By contrast, Revers-Mode Differentiation (RMD) perfectly scales to problems with millions of hyperparameters, but it requires the full inner optimization trajectory of model parameters to be saved, which is computationally costly. Moreover, RMD suffers from gradient vanishing or explosion Antoniou et al. (2018), which leads to training instability. Truncation of the optimization trajectory was proposed to alleviate high memory consumption Shaban et al. (2019) while calculating an approximate hypergradient. However, this approach suffers from short horizon bias Wu et al. (2018). Following Micaelli & Storkey (2020), we define greediness as finding the optimal hyperparameters on a local scale, rather than on a global scale.

Alternatively, an implicit differentiation may be used to compute the hypergradient Lorraine et al. (2020); Luketina et al. (2016); Pedregosa (2016). This approach mitigates the need for unrolling, but it heavily relies on Implicit Function Theorem, which requires the convergence of the inner optimization Grazzi et al. (2020); Blondel et al. (2022). The challenge of this family of methods is computing inverse Hessian-vector product. This computation may be approximated with Neumann series Lorraine et al. (2020) or conjugate gradients Pedregosa (2016).

In this paper, we propose an alternative approach to hypergradient computation. We generalize the method from Luketina et al. (2016). Namely, the proposed approach resolves the following issues simultaneously: short horizon bias, high memory requirements, applicability to large-scale problems with millions of hyperparameters, and independence of inner optimization convergence. Overall, our contributions are as follows:

1. we introduce a procedure that aggregates the greedy gradients calculated at each iteration of the inner objective, which satisfies the requirements above.

2. We provide a theoretical analysis of the proposed approach. Under some assumptions, a sufficient descent condition holds.

3. We empirically prove the effectiveness of the proposed approach on a Meta-Learning task.

The rest of the paper is organized as follows: In Section 2, we briefly review works related to the proposed method. We provide some background information about hypergradient computation in 3. In Section 4, we present the formal problem statement and describe our hyperparameter optimization approach. Section 4.2 shows how our method can be viewed as an extension of the $T1 - T2$ method, while Section 4.3 provides a comprehensive analysis of the proposed approach. In Section 5, we demonstrate the effectiveness of our method through a series of experiments. Finally, we discuss potential directions for future research in Section 6.

## 2 RELATED WORK

**Gradient-Based Hyperaprameter Optimization**. Differentiation through optimization Domke (2012) was successfully applied to hyperparameter optimization at a large-scale Maclaurin et al. (2015). The unrolled differentiation could be categorized into Forward-Mode and Reverse-Mode differentiation Franceschi et al. (2017). The former one suits best for the cases when a handful of hyperparameters is needed to be optimized Micaelli & Storkey (2020), for instance, learning rate and weight dacay. The latter is suitable for the setup with millions of hyperparameters while sacrificing the memory consumption when the number of inner optimization steps increases, except for the cases when SGD with momentum is used Maclaurin et al. (2015). Additionally, truncated unrolled differentiation Shaban et al. (2019) introduces a trade-off between computational complexity and hypergradient accuracy. However, computations done on truncated trajectories suffer from short horizon bias Wu et al. (2018).

Alternatively, impicit differentiation, inspired by the Implicit Function Theorem (IFT), is used to compute the hypergradient Bengio (2000); Lorraine et al. (2020); Pedregosa (2016); Luketina et al.

Table 1: A comparison of gradient-based methods for hyperparameter optmization is presented: Forward-Mode and Reverse-Mode differentiation Franceschi et al. (2017), Implicit Function Theorem Lorraine et al. (2020), $T1 - T2$ Luketina et al. (2016) and the proposed approach. In this context, $P$ refers to the number of model parameters and $H$ denotes the number of hyperparameters. Furthermore, $K$ denotes the number of terms in the approximation using the Neumann series.

| | RMD | FMD | IFT | T1 − T2 | Ours |
|---|---|---|---|---|---|
| Long Horizon | Yes | Yes | Yes | No | Yes |
| Scalable to large amount of hyperparameters | Yes | No | Yes | Yes | Yes |
| Space Complexity | $O(PT)$ | $O(PH)$ | $O(P+H)$ | $O(P+H)$ | $O(P+H)$ |
| Time complexity | $O(T)$ | $O(HT)$ | $O(K)$ | $O(1)$ | $O(T)$ |
| No inner optimality | Yes | Yes | No | Yes | Yes |

(2016). In Bengio (2000) an exact inverse Hessian is computed, which is computationally intractable in huge-scale scenario with millions of model parameters. To sidestep this issue, an approximation is needed. Specifically, the Neumann series approximation Lorraine et al. (2020), conjugate gradients Pedregosa (2016), GMRES Blondel et al. (2022) for solving linear systems, Nyström method Hataya & Yamada (2023), and Broyden's method Hao et al. (2022). The major limitation is that the near-optimality of the inner optimization is crucial for accurate approximation of the true hypergradient Grazzi et al. (2020); Blondel et al. (2022). Moreover, the method is inapplicable to tackling the optimizer hyperaprameters such as learning rate.

We summarize the comparison of described approaches in Table 1.

**Meta-Learning**. Another fundamental application of bilevel optimization is meta-learning Schmidhuber (1987) (or learning to learn). It aims to train a model that generalizes well over the distribution of tasks Ravi & Larochelle (2016). In the context of gradient-based model-agnostic meta-learning Finn et al. (2017), the task is to learn an initialization of model parameters such that gradient-based fine-tuning shows good generalization. MAML optimization successfully inherts the methods for hypergradient computation. Specifically, Li et al. (2018) successfully employed Luketina et al. (2016), Rajeswaran et al. (2019) used implicit differentiation with conjugate gradient algorithm.

## 3 BACKGROUND

In this section we introduce a derivation of an exact hypergradient computation.

Given a vector of model parameters $\mathbf{w} \in \mathbb{R}^P$ and a vector of hyperparameters $\boldsymbol{\alpha} \in \mathbb{R}^H$. The dynamic of model parameters $\{\mathbf{w}_t\}_{t=0}^T$ for some $T \in \mathbb{N}$ and some $\boldsymbol{\alpha}$ is defined as follows $\mathbf{w}_{t+1} = \boldsymbol{\Phi}(\mathbf{w}_t, \boldsymbol{\alpha})$, where $\boldsymbol{\Phi}(.,.)$ is a smooth mapping. For instance, a vanilla gradient descent with stepsize $\eta > 0$ could be written as $\boldsymbol{\Phi}(\mathbf{w}_t, \boldsymbol{\alpha}) = \mathbf{w}_t - \eta \nabla_{\mathbf{w}} \mathcal{L}_{\text{train}}(\mathbf{w}_t, \boldsymbol{\alpha})$, where $\mathcal{L}_{\text{train}}$ is a training loss function. Given also a differentiable validation loss function $\mathcal{L}_{\text{val}}(\mathbf{w}, \boldsymbol{\alpha})$. Under the notations above we formulate a hyperparameter optimization problem as follows:

$$\boldsymbol{\alpha}^* - \arg \min_{\boldsymbol{\alpha} \in \mathbb{R}^H} \mathcal{L}_{\text{val}}(\mathbf{w}_T, \boldsymbol{\alpha}), \tag{1}$$

$$\text{s.t.} \quad \mathbf{w}_t = \boldsymbol{\Phi}(\mathbf{w}_{t-1}, \boldsymbol{\alpha}), \quad t \in \overline{1, T}. \tag{2}$$

Now the goal is to derive a hypergdadient $d_{\boldsymbol{\alpha}} \mathcal{L}_{\text{val}}(\mathbf{w}_T, \boldsymbol{\alpha})$, viewing $\mathbf{w}_T$ as a function of $\boldsymbol{\alpha}$:

$$d_{\boldsymbol{\alpha}} \mathcal{L}_{\text{val}}(\mathbf{w}_T, \boldsymbol{\alpha}) = \nabla_{\boldsymbol{\alpha}} \mathcal{L}_{\text{val}}(\mathbf{w}_T, \boldsymbol{\alpha}) + \nabla_{\mathbf{w}_T} \mathcal{L}_{\text{val}}(\mathbf{w}_T, \boldsymbol{\alpha}) \frac{d\mathbf{w}_T}{d\boldsymbol{\alpha}}. \tag{3}$$

Here $\nabla_{\boldsymbol{\alpha}} \mathcal{L}_{\text{val}}(\mathbf{w}_T, \boldsymbol{\alpha})$ is a row-vector. The chain rule suggests that $d\mathbf{w}_T / d\boldsymbol{\alpha}$ is computed in the following way Franceschi et al. (2017):

$$\frac{d\mathbf{w}_T}{d\boldsymbol{\alpha}} = \sum_{t=1}^{T}\left(\prod_{k=t+1}^{T}\mathbf{A}_k\right)\mathbf{B}_t, \quad \mathbf{A}_k = \frac{\partial\boldsymbol{\Phi}(\mathbf{w}_{k-1},\boldsymbol{\alpha})}{\partial\mathbf{w}_{k-1}}, \quad \mathbf{B}_t = \frac{\partial\boldsymbol{\Phi}(\mathbf{w}_{t-1},\boldsymbol{\alpha})}{\partial\boldsymbol{\alpha}}. \tag{4}$$

Therefore, the hypergradient is calculated as follows:

$$d_{\boldsymbol{\alpha}}\mathcal{L}_{\text{val}}(\mathbf{w}_T,\boldsymbol{\alpha}) = \nabla_{\boldsymbol{\alpha}}\mathcal{L}_{\text{val}}(\mathbf{w}_T,\boldsymbol{\alpha}) + \sum_{t=1}^{T}\nabla_{\mathbf{w}_T}\mathcal{L}_{\text{val}}(\mathbf{w}_T,\boldsymbol{\alpha})\left(\prod_{k=t+1}^{T}\mathbf{A}_k\right)\mathbf{B}_t. \tag{5}$$

The computation of equation 4 could be implemented with a Reverse-Mode Differentiation (RMD) or Forward-Mode Differentiation (FMD) Franceschi et al. (2017). However, the aforementioned method is computationally expensive in terms of either latency (FMD) or memory (RMD). Note that RMD may not need to store the trajectory $\mathbf{w}_1, \ldots, \mathbf{w}_T$ in case of SGD with momentum. However, this would require $2T - 1$ Jacobian-vector products (JVPs), which is computationally demanding. So, we develop the method that performs only $T$ JVPs for the hypergradient computation.

## 4 THE METHOD

### 4.1 HYPERGRADIENT APPROXIMATION

In this section we introduce a computationally efficient approximation to equation 5. Specifically, consider the $t$-th step of the inner optimization. The challenge is that the computation of $\prod_{k=t+1}^{T}\mathbf{A}_k$ requires the tail of the trajectory $\mathbf{w}_t, \ldots, \mathbf{w}_T$. To this end, we introduce an approximation of the product with $\gamma^{T-t}$, where $\gamma \in (0, 1]$. We motivate the choice of $\gamma$ by the fact that $(1-\eta L)\mathbf{I} \preceq \mathbf{A}_k \preceq \mathbf{I}$ if $\mathcal{L}_{\text{train}}(.,\boldsymbol{\alpha})$ is $L$-smooth and convex for any $\boldsymbol{\alpha} \in \mathbb{R}^H$. Indeed, if we assume that $\boldsymbol{\Phi}(.,.)$ is a vanilla gradient descent, then $\mathbf{A}_k = \mathbf{I} - \eta\nabla^2_{\mathbf{w}_{k-1}}\mathcal{L}_{\text{train}}(\mathbf{w}_{k-1},\boldsymbol{\alpha})$. Due to the convexity and $L$-smoothness of $\mathcal{L}_{\text{train}}(.,\boldsymbol{\alpha})$ we conclude that $\mathbf{0} \preceq \nabla^2_{\mathbf{w}_{k-1}}\mathcal{L}_{\text{train}}(\mathbf{w}_{k-1},\boldsymbol{\alpha}) \preceq L\mathbf{I}$. So, choosing the step size $\eta \leq L^{-1}$, we conclude that the spectrum of $\mathbf{A}_k$ is bounded between $0$ and $1$ for any choice of $\boldsymbol{\alpha}$ and $k$. Additionally, we replace the gradient of the validation loss $\nabla_{\mathbf{w}_T}\mathcal{L}_{\text{val}}(\mathbf{w}_T,\boldsymbol{\alpha})$ with the gradient from the current iteration $\nabla_{\mathbf{w}_t}\mathcal{L}_{\text{val}}(\mathbf{w}_t,\boldsymbol{\alpha})$ due to the same reason. Write down the proposed approximation:

$$\hat{d}_{\boldsymbol{\alpha}}\mathcal{L}_{\text{val}}(\mathbf{w}_T,\boldsymbol{\alpha};\gamma) = \nabla_{\boldsymbol{\alpha}}\mathcal{L}_{\text{val}}(\mathbf{w}_T,\boldsymbol{\alpha}) + \sum_{t=1}^{T}\gamma^{T-t}\nabla_{\mathbf{w}_t}\mathcal{L}_{\text{val}}(\mathbf{w}_t,\boldsymbol{\alpha})\mathbf{B}_t. \tag{6}$$

Note that the intuition from equation 6 was previously used in Lee et al. (2021). However, it was used as an intermediate step in the reasoning. Moreover, the approximation of the gradient of the validation loss function w.r.t. model parameters was not considered. Figure shows a schematic overview of the propsed approach.

### 4.2 GENERALIZATION OF $T1 - T2$

Note that the proposed hypergradient computation equation 6 is a generalization of $T1 - T2$ hypergradient Luketina et al. (2016) when $\gamma$ tends to zero. Below we formulate a formal statement.

**Proposition 4.1.** *Let $\hat{d}_{\boldsymbol{\alpha}}(\mathbf{w}_T,\boldsymbol{\alpha};\gamma)$ be the hypergradient defined in equation 6. Then, the following holds:*

$$\lim_{\gamma \to 0^+}\hat{d}_{\boldsymbol{\alpha}}(\mathbf{w}_T,\boldsymbol{\alpha};\gamma) = \nabla_{\boldsymbol{\alpha}}\mathcal{L}_{\text{val}}(\mathbf{w}_T,\boldsymbol{\alpha}) + \nabla_{\mathbf{w}_T}\mathcal{L}_{\text{val}}(\mathbf{w}_T,\boldsymbol{\alpha})\mathbf{B}_T. \tag{7}$$

Here the right hand side of *equation* 7 is the hypergradient of in $T1 - T2$ Luketina et al. (2016). The result given in Proposition 4.1 suggest that $T1 - T2$ hypergradient is a special case of the proposed one. Additionally, it could be clearly seen that the proposed hypergradient computation is conditioned on the whole trajectory of model parameters. We argue that this approach does not suffer from a short-horizon bias problem Wu et al. (2018).

### 4.3 Descent Direction Analysis

Here we discuss the quality of the proposed hypergradient approximation equation 6. We show that the sufficient descent condition holds under some assumptions. Inspired by Shaban et al. (2019); Ghadimi & Wang (2018), we first formulate a standard set of assumptions.

**Assumption 4.2.** *Suppose that the following assumptions on the functions $\mathcal{L}_{\text{train}}(.,.)$, $\mathcal{L}_{\text{val}}(.,.)$, and the optimization operator $\mathbf{\Phi}(.,.)$ are satisfied:*

1. *$\mathcal{L}_{\text{val}}(., \boldsymbol{\alpha})$ is $L$-smooth and $\mu$-strongly convex for any $\boldsymbol{\alpha}$.*

2. *$\frac{\partial \mathbf{\Phi}(., \boldsymbol{\alpha})}{\partial \boldsymbol{\alpha}}$ is $C_B$-Lipschitz for any $\boldsymbol{\alpha}$.*

3. *$\|\frac{\partial \mathbf{\Phi}(\mathbf{w}, \boldsymbol{\alpha})}{\partial \boldsymbol{\alpha}}\|_{op} \leq B$ for any pair $(\mathbf{w}, \boldsymbol{\alpha})$ for some $B \geq 0$.*

4. *$\mathbf{w}$ belongs to a bounded convex set with diameter $D < \infty$.*

5. *$\mathbf{\Phi}(\mathbf{w}, \boldsymbol{\alpha}) = \mathbf{w} - \eta \nabla_{\mathbf{w}} \mathcal{L}_{\text{train}}(\mathbf{w}, \boldsymbol{\alpha})$ for some $\eta \geq 0$.*

Second, we formulate and justify specific assumptions.

**Assumption 4.3.** *Suppose that the following holds for $\mathcal{L}_{\text{train}}(.,.)$ and $\mathcal{L}_{\text{val}}(.,.)$:*

1. *$\nabla^2_{\mathbf{w}} \mathcal{L}_{\text{train}}(., \boldsymbol{\alpha}) = \mathbf{I}$ for any $\boldsymbol{\alpha}$. Note that this assumption does not hold in practice. However, Luketina et al. (2016) argues that batch normalization Ioffe & Szegedy (2015) forces the Hessian to be close to the identity matrix.*

2. *$\nabla_{\boldsymbol{\alpha}} \mathcal{L}_{\text{val}}(\mathbf{w}, \boldsymbol{\alpha}) = \mathbf{0}$ for any $\mathbf{w}$. This assumption is typical for hyperparameter optimization and data hypercleaning Franceschi et al. (2017).*

3. *$\mathbf{B}_t \mathbf{B}_t^\top \succeq \kappa \mathbf{I}$ for some $\kappa > 0$. We note that the assumption that $\mathbf{B}_t$ is a full-rank matrix was used in Shaban et al. (2019). However, we impose more strict assumption to simplify the proofs.*

4. *Define $\mathbf{w}_\infty := \arg\min_{\mathbf{w}} \mathcal{L}_{\text{train}}(\mathbf{w}, \boldsymbol{\alpha})$, $\mathbf{w}_2^* := \arg\min_{\mathbf{w}} \mathcal{L}_{\text{val}}(\mathbf{w}, \boldsymbol{\alpha})$. Assume that $\|\mathbf{w}_\infty - \mathbf{w}_2^*\| \geq 2De^{-\mu\eta T} + \delta$, for some $\delta > 0$. Also assume that $\nabla_{\mathbf{w}_2^*} \mathcal{L}_{\text{val}}(\mathbf{w}_2^*, \boldsymbol{\alpha}) = \mathbf{0}$ for any $\boldsymbol{\alpha}$. Intuitively, this requirements asserts that an overfitting takes place, and the minimum is reached in the interior of the feasible set.*

**Lemma 4.4.** *(Shaban et al. (2019)) In the assumptions above 4.2, 4.3, the sequence $\{\mathbf{w}_t\}_{t \geq 0}$ satisfies:*

$$\|\mathbf{w}_t - \mathbf{w}_\infty\|_2 \leq \|\mathbf{w}_0 - \mathbf{w}_\infty\|_2 e^{-\eta t}. \tag{8}$$

**Lemma 4.5.** *Let the assumptions 4.2, 4.3 hold. Then the following is true:*

$$\|\nabla_{\mathbf{w}_T} \mathcal{L}_{\text{val}}(\mathbf{w}_T, \boldsymbol{\alpha})\|_2 \geq \mu\delta. \tag{9}$$

The following theorem guarantees that the proposed hypergradient is a sufficient descent direction.

**Theorem 4.6.** *Suppose that $\gamma = 1 - \eta \in (0, 1)$. Then, under the assumptions above 4.2, 4.3, there exists a sufficiently large $T$ and a universal constant $c > 0$ such that:*

$$d_{\boldsymbol{\alpha}} \mathcal{L}_{\text{val}}(\mathbf{w}_T, \boldsymbol{\alpha}) \hat{d}_{\boldsymbol{\alpha}} \mathcal{L}_{\text{val}}(\mathbf{w}_T, \boldsymbol{\alpha}; \gamma)^\top \geq c\|d_{\boldsymbol{\alpha}} \mathcal{L}_{\text{val}}(\mathbf{w}_T, \boldsymbol{\alpha})\|_2^2.$$

## 5 Experiments

In this section we present numerical experiments that validate the effectiveness and efficiency of the proposed approach. Upon acceptance, we will make the source codes available.

## 5.1 BASELINES

For comparison, we consider the following list of baselines that are efficient in terms of space and latency:

- **T1 − T2** Luketina et al. (2016). The method performs an unrolled differentiation using only the last step of inner optimization, so it performs a JVP.

- **IFT** Lorraine et al. (2020). The method combines the implicit function theorem (IFT) with efficient approximations of the inverse Hessian. The number of JVPs is controlled by the number of terms taken from the Neumann series.

- **FO**. The method uses only the first-order gradient from equation 5, namely $\nabla_{\boldsymbol{\alpha}} \mathcal{L}_{\text{val}}(\mathbf{w}_T, \boldsymbol{\alpha})$. Note that it is not applicable for tasks for which the outer objective does not depend explicitly on the vector of hyperparameters $\boldsymbol{\alpha}$.

- **Full**. The method computes the true hypergradient defined in equation 5.

## 5.2 TOY PROBLEM

Following Shaban et al. (2019), we formulate a toy bilevel problem with the following objectives:

$$\mathcal{L}_{\text{val}}(\mathbf{w}, \boldsymbol{\alpha}) = \|\mathbf{w}\|_2^2 + 10\|\sin(\mathbf{w})\|_2^2, \tag{10}$$

$$\mathcal{L}_{\text{train}}(\mathbf{w}, \boldsymbol{\alpha}) = \frac{1}{2}(\mathbf{w} - \boldsymbol{\alpha})^\top \mathbf{G}(\mathbf{w} - \boldsymbol{\alpha}), \tag{11}$$

where $\mathbf{w} \in \mathbb{R}^2$ and $\mathbf{G} = \text{diag}(1, \frac{1}{2})$. We solve both inner and outer problems using SGD with a learning rate of 0.1 and without momentum. The initial parameters are $\mathbf{w}_0 = (2, 2)$ and hyperparameters $\boldsymbol{\alpha}_0 = (1, 1)$.

We report a validation loss on the outer optimization steps for the proposed approach and all the baselines described in Section 5.1, except for FO, since the outer objective does not depend on the hyperparameters explicitly. Additionally, we report the cosine similarity between the true hypergradient and the approximation for each outer iteration. The results for horizon lengths $T \in \{5, 20\}$ are presented in Figure 2.

It could be clearly seen from Figure 2 that the proposed approach with $\gamma = 0.9$ achieves the best validation performance regardless the horizon length $T$, outperforming all the baselines. Moreover, the cosine similarity plots indicate that the proposed generalization outperforms the vanilla $T1 - T2$. However, IFT approach performs on par with ours with $\gamma = 0.9$. Interestingly, there is a non-monotonic behaviour observed in the proposed method with $T = 10$ and $\gamma = 0.9$. We leave this phenomenon for future research.

## 5.3 DATA HYPER-CLEANING

Following Franceschi et al. (2017), the task is formulated as follows. Given a training dataset $\mathfrak{D}_{\text{train}} = \{(\mathbf{x}_i, y_i)\}_{i=1}^{n_{\text{train}}}$, where $\mathbf{x}_i$ is an object and $y_i$ is a class label. Similary, define a validation dataset $\mathfrak{D}_{\text{val}} = \{(\mathbf{x}_i, y_i)\}_{i=1}^{n_{\text{val}}}$. We assume that the labels of the training dataset are corrupted. More precisely, the label is replaced by a random class with probability $p_{\text{noise}}$. To mitigate the influence of noisy labels we introduce a vector of weights for each training object $\boldsymbol{\alpha} \in \mathbb{R}^{n_{\text{train}}}$. The task is to find a vector such that the model trained on the reweighted samples achives the optimal validation performance on clan data. Given model parameters $\mathbf{w}$. The training loss function is $\mathcal{L}_{\text{train}}(\mathbf{w}, \boldsymbol{\alpha}) = \sum_{(\mathbf{x}_i, y_i) \in \mathfrak{D}_{\text{train}}} \sigma(\alpha_i)\ell(\mathbf{w}, \mathbf{x}_i, y_i)$, where $\sigma(.)$ is a sigmoid function, $\ell(.)$ is a cross-entropy loss function for the training pair $(\mathbf{x}_i, y_i)$. The validation loss function is $\mathcal{L}_{\text{val}}(\mathbf{w}, \boldsymbol{\alpha}) = \sum_{(\mathbf{x}_i, y_i) \in \mathfrak{D}_{\text{val}}} \ell(\mathbf{w}, \mathbf{x}_i, y_i)$.

We run the experiment on MNIST LeCun et al. (1998) and Fashion-MNIST Xiao (2017) datasets. We randomly select a subset of 1000 instances from the training split for the inner objective. As for the clean validation data, we take the whole test split. The inner optimization is done in full-batch manner using SGD with a learning rate of $10^{-1}$ and momentum 0.9, while the outer problem is optimized with Adam with a learning rate of $10^{-1}$. As for a model, we used a 3-layer convolutional network with 8 channels. We set the number of inner steps to $T = 10$ and the number of outer updates to 200.

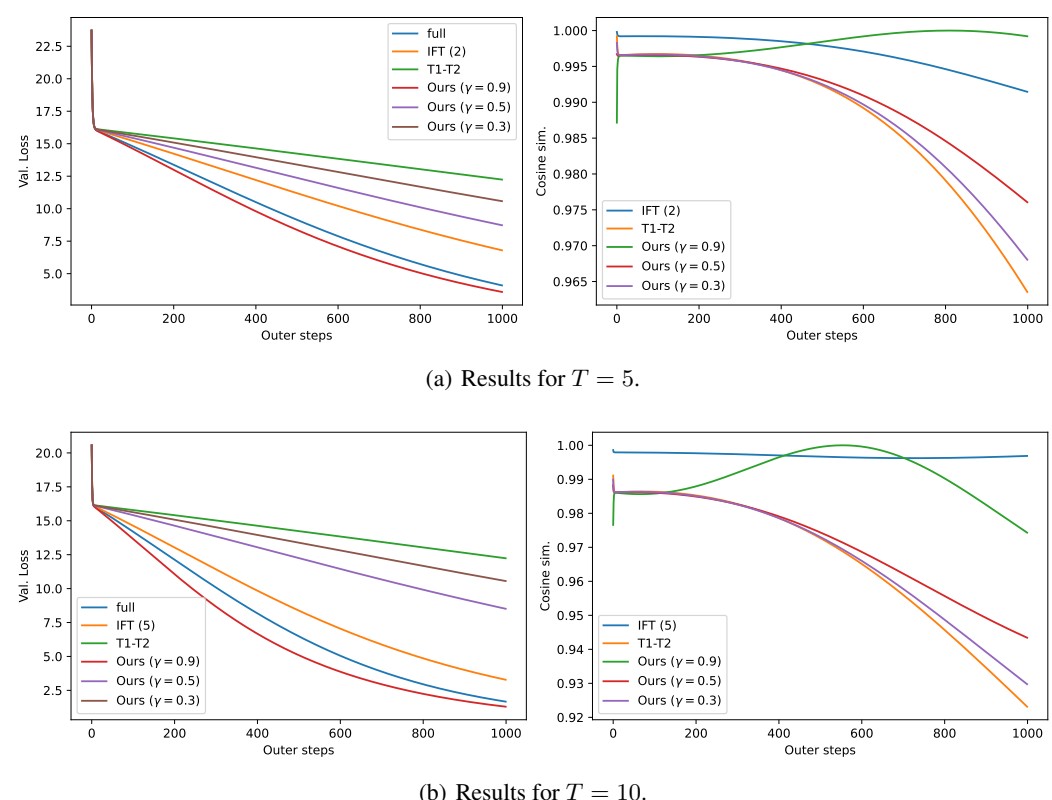

(a) Results for $T = 5$.

(b) Results for $T = 10$.

Figure 2: Results for a toy experiment. Validation loss and cosine similarity with the true hypergradient is presented.

| Method | #JVPs | MNIST (0.3) | MNIST (0.5) | F-MNIST (0.3) | F-MNIST (0.5) |
|---|---|---|---|---|---|
| w/o HPO | 0 | $70.46 \pm 5.47$ | $51.52 \pm 1.61$ | $59.46 \pm 4.12$ | $53.99 \pm 6.88$ |
| $T1 - T2$ | 1 | $66.93 \pm 1.2$ | $51.86 \pm 2.77$ | $62.07 \pm 4.41$ | $51.40 \pm 5.53$ |
| IFT (5) | 11 | $71.97 \pm 4.54$ | $54.14 \pm 6.66$ | $60.63 \pm 5.03$ | $47.68 \pm 1.51$ |
| Ours ($\gamma = 0.9$) | 10 | $\mathbf{87.06 \pm 0.77}$ | $\mathbf{77.73 \pm 1.70}$ | $\mathbf{72.57 \pm 1.09}$ | $\mathbf{66.21 \pm 1.50}$ |

Table 2: The results for data hyper-cleaning experiment. Validation accuracy is reported. The value of $p_{\text{noise}}$ is presented in parenthesis.

We tune $\gamma$ within the set $\{0.9, 0.99, 0.999\}$ and select the best-performing value for each $p_{\text{noise}}$. The experiments have demonstrated that $\gamma = 0.9$ performs uniformly well.

We report the validation accuracy and 95% confidence interval for five trials of the compared baselines and the proposed method in Table 2 for different values of $p_{\text{noise}}$. We also report the number of JVPs. To illustrate the effect from hyperparameter optimization, the metrics for the baseline without hyper-cleaning are reported, i.e. $\boldsymbol{\alpha} = \mathbf{0}$. The results suggest that the proposed method outperforms the baselines in terms of validation accuracy, having comparable computational cost.

## 5.4 GRADIENT-BASED META-LEARNING

We consider gradient-base Meta-Learning task for few-shot image classification task Finn et al. (2017) in a $K$-shot $m$-way setting. As for the model, we consider a 6-layer convolutional network with 32 channels. Inspired by Flennerhag et al. (2019), we treat the 2-nd, 4-th and 6-th layer as a high-dimensional hyperparameters that are not optimized in the inner loop. We conduct the experiment on Omniglot Lake et al. (2011) dataset, downsampled to $28 \times 28$. We leave $20\%$ of the classes for meta-validation split and the validation dataset for each task consists of 10 samples for each class.

| Method | #JVPs | 2-way, 1-shot | 3-way, 1-shot | 5-way, 1-shot | 10-way, 1-shot |
|---|---|---|---|---|---|
| FO | 0 | $87.31 \pm 1.79$ | $78.22 \pm 1.04$ | $69.71 \pm 0.5$ | $62.41 \pm 1.95$ |
| $T1 - T2$ | 1 | $89.4 \pm 0.36$ | $78.25 \pm 1.03$ | $66.7 \pm 3.09$ | $52.23 \pm 1.63$ |
| IFT (2) | 5 | $87.1 \pm 2.52$ | $79.36 \pm 2.31$ | $72.57 \pm 2.14$ | $53.62 \pm 5.92$ |
| Ours ($\gamma = 0.9$) | 5 | $\mathbf{93.5 \pm 0.34}$ | $\mathbf{86.83 \pm 0.78}$ | $\mathbf{80.46 \pm 1.08}$ | $\mathbf{66.92 \pm 2.04}$ |

Table 3: Few-shot accuracy on the meta-learning task.

The inner optimization is done using SGD with a learning rate of $10^{-1}$ and momentum 0.9, while the outer problem is optimized with Adam with a learning rate of $10^{-3}$. The number of outer steps is set to $2 \cdot 10^3$ and the horizon length $T$ is set to 5. The inner optimization is done in a full-batch manner. We tune $\gamma$ for the proposed approach within the set $\{0.9, 0.99, 0.999\}$ and select the best-performing value for each task using the meta-validation split. Interestingly, $\gamma = 0.9$ performs remarkably well irrespective of the task.

The accuracy on meta-validation split is presented in Table 3 for different few-shot scenarios, along with the number of JVPs. We report the mean and a 95% confidence interval based on 5 trials using different random seeds . It could be clearly seen that the proposed approach shows substantial improvements over the baselines in terms of accuracy on the meta-validation split.

## 6 FUTURE WORK AND EXTENSIONS

**Hyperparameter $\gamma$ estimation.** One of the future work directions is an exploration of the optimal value of the hyperparameter $\gamma$. While we have not yet conducted a comprehensive analysis, current experiments suggest that $\gamma = 0.9$ offers strong performance, comparable or superior to baseline methods. A straightforward approach to tuning this hyperparameter is through grid or random search. However, theoretical framework proposed in Section 4 establishes a relation between $\gamma$ and matrices of parameter gradients $\mathbf{A}_k$. This fact can be used for potential analytical methods to derive its optimal value.

**Extension to Other Optimization Algorithms.** The proposed method can be viewed as an extension of the $T1 - T2$ method, leveraging a longer optimization horizon and the inclusion of momentum. The momentum term establishes connections with other optimization algorithms, and future extensions could incorporate advanced optimization techniques such as adaptive moment estimation Kingma & Ba (2015). Additionally, a promising direction for further research is to explore neural network-based optimization methods, as demonstrated in Andrychowicz et al. (2016), which could potentially improve the adaptability of the proposed method.

**Validation Loss Surface and Horizon Length in Hyperparameter Optimization.** Our experiments demonstrate that, across multiple tasks, the proposed method outperforms more sophisticated approaches such as the IFT method, despite its simplicity in both computational and implementation aspects. This raises a question: how complex is the underlying hyperparameter optimization problem, and do we truly require accurate hyperparameter gradient approximations over a long horizon? While several studies theoretically explore hyperparameter optimization with long horizon Micaelli & Storkey (2020); Wu et al. (2018), the complexity of the validation loss surface for real-world problems remains an open question and needs to be investigated.

## 7 CONCLUSION

The paper presents an approximation of the true hypergradient for gradient-based bilevel optimization that avoids the high memory cost and short horizon bias. Additionally, the method does not require the assumption of convergence to an optimal solution for the inner optimization. The proposed method exploits an aggregation of greedy gradients calculated at each step of the inner trajectory. Our theoretical findings suggest that the approximation satisfies the sufficient descent condition. Empirically, the introduced method outperforms the baselines in terms of validation accuracy, having

comparable computational costs. One promising direction for future research is to investigate more accurate Hessian approximations.

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

# A  APPENDIX

## A.1  PROOFS FOR THE RESULTS IN THE MAIN TEXT

*Proof of Proposition 4.1.* First, using the definition of $\hat{d}_{\boldsymbol{\alpha}}(\mathbf{w}_T, \boldsymbol{\alpha}; \gamma)$ from equation 6, we conclude that:

$$\hat{d}_{\boldsymbol{\alpha}}(\mathbf{w}_T, \boldsymbol{\alpha}; \gamma) = \nabla_{\boldsymbol{\alpha}} \mathcal{L}_{\text{val}}(\mathbf{w}_T, \boldsymbol{\alpha}) + \nabla_{\mathbf{w}_T} \mathcal{L}_{\text{val}}(\mathbf{w}_T, \boldsymbol{\alpha}) \mathbf{B}_T + \gamma \sum_{t=1}^{T-1} \gamma^{T-t-1} \nabla_{\mathbf{w}_t} \mathcal{L}_{\text{val}}(\mathbf{w}_t, \boldsymbol{\alpha}) \mathbf{B}_t.$$

Second, note that the last term tends to zero:

$$\lim_{\gamma \to 0^+} \gamma \sum_{t=1}^{T-1} \gamma^{T-t-1} \nabla_{\mathbf{w}_t} \mathcal{L}_{\text{val}}(\mathbf{w}_t, \boldsymbol{\alpha}) \mathbf{B}_t = \mathbf{0}.$$

The combination of the above two steps completes the proof. $\qquad \square$

*Proof of Lemma 4.5.* First, use the Polyak-Lojasiewicz condition, since $\mathcal{L}_{\text{val}}(.,.)$ is $\mu$-strongly convex in the first argument due to 4.2. Second, use the strong convexity of $\mathcal{L}_{\text{val}}(., \boldsymbol{\alpha})$ according to 4.2. Third, use Lemma 4.4 for $\mathbf{w}_T$, and finally the overfitting condition from 4.3:

$$\|\nabla_{\mathbf{w}_T} \mathcal{L}_{\text{val}}(\mathbf{w}_T, \boldsymbol{\alpha})\|_2^2 \overset{4.2(1)}{\geq} 2\mu(\mathcal{L}_{\text{val}}(\mathbf{w}_T, \boldsymbol{\alpha}) - \mathcal{L}_{\text{val}}(\mathbf{w}_2^*, \boldsymbol{\alpha}))$$

$$\overset{4.2(1)}{\geq} \mu^2 \|\mathbf{w}_T - \mathbf{w}_2^*\|^2$$

$$\geq \mu^2 (\|\mathbf{w}_T - \mathbf{w}_\infty\|_2^2 + \|\mathbf{w}_2^* - \mathbf{w}_\infty\|_2^2 - 2\|\mathbf{w}_T - \mathbf{w}_\infty\|_2 \cdot \|\mathbf{w}_2^* - \mathbf{w}_\infty\|_2)$$

$$\overset{4.4}{\geq} \mu^2 (\|\mathbf{w}_2^* - \mathbf{w}_\infty\|_2 - 2De^{-\mu\eta T})\|\mathbf{w}_2^* - \mathbf{w}_\infty\|_2$$

$$\overset{4.3(4)}{\geq} \mu^2 \delta^2.$$

$\qquad \square$

*Proof of Theorem 4.6.* Define $\mathbf{g}_j := \nabla_{\mathbf{w}_j} \mathcal{L}_{\text{val}}(\mathbf{w}_j, \boldsymbol{\alpha})$ for $j \in \{1, \ldots, T\}$. Write down the dot product taking into account that $\prod_{k=t+1}^{T} \mathbf{A}_k = (1-\eta)^{T-t}$ according to 4.3(1):

$$d_{\boldsymbol{\alpha}} \mathcal{L}_{\text{val}}(\mathbf{w}_T, \boldsymbol{\alpha}) \hat{d}_{\boldsymbol{\alpha}} \mathcal{L}_{\text{val}}(\mathbf{w}_T, \boldsymbol{\alpha}; \gamma)^\top = \sum_{j=1}^{T} \sum_{t=1}^{T} (1-\eta)^{2T-t-j} \nabla_{\mathbf{w}_T} \mathcal{L}_{\text{val}}(\mathbf{w}_T, \boldsymbol{\alpha}) \mathbf{B}_t \mathbf{B}_j^\top \nabla_{\mathbf{w}_j} \mathcal{L}_{\text{val}}(\mathbf{w}_j, \boldsymbol{\alpha})^\top$$

$$= \sum_{j=1}^{T} \sum_{t=1}^{T} (1-\eta)^{2T-j-t} \mathbf{g}_T \mathbf{B}_t \mathbf{B}_j^\top \mathbf{g}_j.$$

Now estimate each term from below

$$\mathbf{g}_T \mathbf{B}_t \mathbf{B}_j^\top \mathbf{g}_j = \mathbf{g}_T \mathbf{B}_t \mathbf{B}_t^\top \mathbf{g}_j + \mathbf{g}_T \mathbf{B}_t (\mathbf{B}_j - \mathbf{B}_t)^\top \mathbf{g}_j$$

$$\overset{4.2(2)}{\geq} \mathbf{g}_T \mathbf{B}_t \mathbf{B}_t^\top \mathbf{g}_j - C_B \|\mathbf{w}_j - \mathbf{w}_t\|_2 \cdot \|\mathbf{g}_j\|_2 \cdot \|\mathbf{g}_T\|_2 \cdot \|\mathbf{B}_t\|_{\mathrm{op}}$$

$$\overset{4.3(4),4.2(3)}{\geq} \mathbf{g}_T \mathbf{B}_t \mathbf{B}_t^\top \mathbf{g}_j - C_B B \|\mathbf{w}_j - \mathbf{w}_t\|_2 \cdot \|\mathbf{g}_j - \nabla_{\mathbf{w}_2^*} \mathcal{L}_{\mathrm{val}}(\mathbf{w}_2^*, \boldsymbol{\alpha})\|_2 \cdot \|\mathbf{g}_T\|_2$$

$$\overset{4.2(1)}{\geq} \mathbf{g}_T \mathbf{B}_t \mathbf{B}_t^\top \mathbf{g}_j - C_B B \|\mathbf{w}_j - \mathbf{w}_t\|_2 \cdot L \|\mathbf{w}_j - \mathbf{w}_2^*\|_2 \cdot \|\mathbf{g}_T\|_2$$

$$\geq \mathbf{g}_T \mathbf{B}_t \mathbf{B}_t^\top \mathbf{g}_j - C_B B L D \|\mathbf{w}_j - \mathbf{w}_\infty + \mathbf{w}_\infty - \mathbf{w}_t\|_2 \|\mathbf{g}_T\|_2$$

$$\overset{equation\ 8}{\geq} \mathbf{g}_T \mathbf{B}_t \mathbf{B}_t^\top \mathbf{g}_j - C_B B L D (\|\mathbf{w}_0 - \mathbf{w}_\infty\|_2 e^{-\eta t} + \|\mathbf{w}_0 - \mathbf{w}_\infty\|_2 e^{-\eta j}) \|\mathbf{g}_T\|_2$$

$$\overset{4.2(4)}{\geq} \mathbf{g}_T \mathbf{B}_t \mathbf{B}_t^\top \mathbf{g}_j - C_B B L D^2 (e^{-\eta t} + e^{-\eta j}) \|\mathbf{g}_T\|_2$$

Now bound $\mathbf{g}_T \mathbf{B}_t \mathbf{B}_t^\top \mathbf{g}_j$ from below:

$$\mathbf{g}_T \mathbf{B}_t \mathbf{B}_t^\top \mathbf{g}_j = \mathbf{g}_T \mathbf{B}_t \mathbf{B}_t^\top \mathbf{g}_T + \mathbf{g}_T \mathbf{B}_t \mathbf{B}_t^\top (\mathbf{g}_j - \mathbf{g}_T)$$

$$\overset{4.2(1)(3)}{\geq} \kappa \|\mathbf{g}_T\|_2^2 - L \|\mathbf{g}_T\|_2 B^2 \|\mathbf{w}_j - \mathbf{w}_T\|_2$$

$$\overset{equation\ 8}{\geq} \kappa \|\mathbf{g}_T\|_2^2 - L \|\mathbf{g}_T\|_2 B^2 \|\mathbf{w}_0 - \mathbf{w}_\infty\|_2 (e^{-\eta T} + e^{-\eta j})$$

$$\overset{4.2(4)}{\geq} \kappa \|\mathbf{g}_T\|_2^2 - L D B^2 \|\mathbf{g}_T\|_2 (e^{-\eta T} + e^{-\eta j}).$$

Combining together the above bounds, we have:

$$\sum_{j=1}^{T} \sum_{t=1}^{T} (1 - \eta)^{2T-j-t} \mathbf{g}_T \mathbf{B}_t \mathbf{B}_j^\top \mathbf{g}_j \geq$$

$$\kappa T^2 \|\mathbf{g}_T\|_2^2 - C_B B L D^2 \|\mathbf{g}_T\|_2 \sum_{j=1}^{T} \sum_{t=1}^{T} [e^{-\eta t} + e^{-\eta j}] - L D B^2 \|\mathbf{g}_T\|_2 (T^2 e^{-\eta T} + T \sum_{j=1}^{T} e^{-\eta j}) \geq$$

$$\kappa T^2 \|\mathbf{g}_T\|_2^2 - 2 C_B B L D^2 \|\mathbf{g}_T\|_2 T (e^\eta - 1)^{-1} - L D B^2 \|\mathbf{g}_T\|_2 (T^2 e^{-\eta T} + T \eta^{-1}) \geq$$

$$\kappa T^2 \|\mathbf{g}_T\|_2^2 - 2 C_B B L D^2 \|\mathbf{g}_T\|_2 T (e^\eta - 1)^{-1} - L D B^2 \|\mathbf{g}_T\|_2 (T^2 e^{-\eta T} + T (e^\eta - 1)^{-1}).$$

Using Lemma 4.5 we make the following statement. Since the first term of the bound is $\Theta(T^2)$ and the second and the third are $\Theta(T)$, then there exists sufficiently large $T$ and a universal constant $c$ such that the expression is bounded from below with $c\|\mathbf{g}_T\|_2^2$ for $\|\mathbf{g}_T\|_2 \geq \mu\delta$. $\qquad\square$

