# OpenReview forum: "Generalized Greedy Gradient-Based Hyperparameter Optimization"
_ICLR.cc/2025/Conference — Submitted to ICLR 2025_

### Official Review · Reviewer_iGrL · 2024-10-29

**Soundness:** 2
**Presentation:** 2
**Contribution:** 2
**Rating:** 5
**Confidence:** 4

**Summary:**

This paper introduces a novel gradient-based approach for bilevel optimization (BLO), specifically targeting hyperparameter optimization in machine learning tasks. Traditional gradient-based methods face challenges due to memory constraints, dependency on convergence, and short-horizon biases. To address these, the authors propose an approximation that accumulates short-horizon gradients at each step of the inner optimization loop. This method aims to balance memory efficiency and accuracy, ensuring the computed hypergradients remain robust without requiring convergence of the inner loop.

**Strengths:**

1. This method shows a new method to solve the hyperparameter optimization. The authors offer a practical and efficient alternative that balances memory requirements and performance accuracy by accumulating short-horizon gradients from each inner optimization step.
2. The author shows that theoretically, the hypergradient used in this paper is a sufficient descent direction.
3. Numerical experiments validate the effectiveness and efficiency of the proposed approach.

**Weaknesses:**

1. The presentation is not good. Authors should check the format of citations. Eqn (1) seems not correct. It should be $\alpha^*=...$ not $\alpha^*-$. In line 200, the number of figure is missing.
2. The contribution of this paper is limited. To me, the main contribution of this paper is it proposed a new method to approximate the Hessian in hyperparagradient. However, this method is used in Lee et al.(2021) .  Can authors show the approximation error bound between the proposed gradient (6) and the exact gradient (5)?
3. The assumptions used are highly restrictive. For example, the author assumes the upper-level objective is strongly convex.  In most cases, the upper-level objective is assumed to be nonconvex and the lower-level objective is strongly convex or satisfies the PL condition.
4. The authors claim the proposed method can solve large-scale problems. However, only small datasets are used in experiments. This is not enough to show the ability to solve large-scale problems and more large-scale datasets are required. In addition, more experiments of different T are required to show the choice of $\gamma$.
5. It is better to compare more recent methods, such as SOBA[1], FDS[2].
[1] Dagréou M, Ablin P, Vaiter S, et al. A framework for bilevel optimization that enables stochastic and global variance reduction algorithms[J]. Advances in Neural Information Processing Systems, 2022, 35: 26698-26710.
[2] Micaelli P, Storkey A J. Gradient-based hyperparameter optimization over long horizons[J]. Advances in Neural Information Processing Systems, 2021, 34: 10798-10809.

**Questions:**

See weakness

---

### Official Review · Reviewer_ZrSS · 2024-11-04

**Soundness:** 2
**Presentation:** 3
**Contribution:** 2
**Rating:** 5
**Confidence:** 3

**Summary:**

The paper introduces a novel gradient-based method for hyperparameter optimization within bilevel optimization frameworks. It addresses the limitations of existing approaches, such as high memory requirements in reverse-mode differentiation and the convergence dependency in implicit differentiation methods. The proposed method accumulates short-horizon approximations from each step of the inner optimization trajectory, aiming to provide a more efficient and effective hypergradient computation.

**Strengths:**

The paper leverages aggregations of the greedy gradients calculated at each iteration to improve the efficiency of gradient-based hyperparameter optimization methods. It also shows by using the proposed method, a sufficient descent condition holds.

**Weaknesses:**

1. The proposed algorithm still requires T Jacobian-vector products, and it scales up with the number of steps. This means that the proposed method probably cannot be scaled to large-scale tasks. Is there any numeric comparison of the time costs or memory costs between the proposed method, Eq.5, and T1-T2?
2. Eq.6 is an approximation of Eq.5. However, no approximation error is given. Only a rough bound of L_train is given: 0 ⪯ ∇2
w_k−1 L_train(w_k−1, α) ⪯ LI. Is it possible to give a more precise approximation error?
3. The comparison baselines and tasks are old and of smaller scales. The comparison baselines come mostly before 2020. The tasks are also not on a large scale, and the largest dataset it uses is FashionMNIST.

**Questions:**

Please see the weakness.

---

### Official Review · Reviewer_q9MC · 2024-11-10

**Soundness:** 2
**Presentation:** 3
**Contribution:** 2
**Rating:** 3
**Confidence:** 3

**Summary:**

This paper introduces an innovative bilevel optimization approach that aims to enhance computational efficiency by approximating the sequential multiplication of core matrices from step k to T. This process incorporates the Lipschitz smooth parameter to ensure stability and convergence within the optimization. The proposed method is designed to handle the inherent challenges of bilevel optimization, which often involve nested structures where the solution to one optimization problem depends on another. By focusing on approximating matrix products efficiently, the method reduces computational overhead and improves scalability for large-scale applications.

Additionally, the authors provide a comprehensive theoretical analysis that substantiates the convergence properties and robustness of their approach. The analysis includes mathematical proofs that outline the conditions under which the method guarantees convergence and maintains the Lipschitz continuity required for smooth optimization landscapes.

To validate their theoretical findings, the paper also presents extensive empirical results. These experiments demonstrate the practical effectiveness of the method across various optimization scenarios, showcasing improvements in solution quality and computational performance compared to existing approaches. The results confirm that the new method achieves a favorable balance between accuracy and efficiency, making it a promising tool for applications involving complex bilevel optimization problems.

**Strengths:**

The paper under review demonstrates several strengths that make it a notable contribution to the field of optimization and machine learning:

    Significance of the Problem: The paper addresses a highly significant problem in optimization, specifically in the realm of bilevel optimization, which has broad implications for various applications such as hyperparameter tuning, meta-learning, and reinforcement learning. Tackling the challenge of approximating matrix multiplication with a focus on maintaining Lipschitz smoothness is vital for ensuring stability and efficiency in complex, nested optimization scenarios. This makes the research both relevant and impactful, adding value to the community seeking practical solutions to computationally intensive optimization problems.

    Originality of the Idea: The proposed approach showcases originality through its novel approximation strategy. By introducing a method that approximates the multiplication of core matrices across a sequence of steps with consideration of the Lipschitz smooth parameter, the authors present a unique solution that bridges gaps left by previous methods. This concept is particularly innovative as it leverages mathematical structures to enhance the efficiency of bilevel optimization, setting the paper apart from standard approaches.

    Theoretical Rigor and Quality: The paper includes rigorous theoretical analysis to support its claims. The authors have detailed proofs that establish the convergence properties and ensure that the Lipschitz continuity is preserved throughout the optimization process. The quality of the theoretical work is underscored by its depth, covering necessary and sufficient conditions for convergence. This enhances the credibility of the proposed method and demonstrates the authors’ thorough understanding of the underlying mathematical principles.

    Empirical Validation: The empirical results provided in the paper further strengthen its contributions by verifying the practical effectiveness of the proposed method. The authors have conducted a comprehensive set of experiments that illustrate the method's superior performance compared to existing techniques. These experiments cover a variety of scenarios, showcasing improvements in accuracy and computational efficiency. The clarity of the empirical section is commendable, as it transparently presents metrics, comparisons, and insights that validate the theoretical expectations. The empirical findings support the claim that the proposed method is a viable and competitive option for tackling bilevel optimization problems in real-world applications.

**Weaknesses:**

Approximation Quality of Core Matrices' Multiplication: One primary concern is the quality of the approximation when performing sequential multiplication of core matrices from step kk to TT while maintaining the Lipschitz smooth parameter. It is crucial to ensure that this approximation does not compromise the integrity and stability of the results, particularly in more complex or high-dimensional scenarios where the error might accumulate significantly. The paper should provide more detailed analysis or bounds on how this approximation impacts overall optimization performance, as this would strengthen the confidence in the method’s robustness and applicability. Without this, readers may question the method's reliability, especially in comparison to more established algorithms.

Moderate Level of Novelty: Although the idea behind the proposed method is interesting, the degree of novelty appears moderate. While it does introduce an innovative approximation strategy, some aspects of the method align closely with existing work in the field. The connection between this approach and previous methods may need clearer delineation to highlight what aspects are genuinely new. This would involve not only positioning the paper within the broader research landscape but also elaborating on what sets it apart. A more thorough discussion of related work and how this method extends, diverges from, or improves upon them would add to the paper's originality.

Theoretical Results Compared to Previous Work: Theoretical results presented in the paper, while comprehensive, appear somewhat hesitant or less robust when compared to those in earlier influential publications. While the authors provide proofs that support the proposed approach, there seems to be an opportunity to strengthen these results by either deepening the mathematical analysis or comparing them more directly to well-established theories. Doing so would enhance the perceived contribution of the paper, making it clearer how these results build upon or surpass the theoretical guarantees of prior studies. This comparative aspect is essential for demonstrating the added value of the new method beyond incremental improvements.

Lack of Comprehensive Reporting on Running Time: Although the paper provides empirical results that compare performance in terms of optimization quality over iterations, it falls short in reporting the actual running time of the proposed method. Including detailed runtime analysis is essential for practitioners who need to balance performance with computational cost, particularly for large-scale problems where running time can be a critical constraint. Without such data, it is challenging to assess the practical feasibility of the method. To address this, a section dedicated to computational efficiency, including comparisons with baseline methods, would be valuable. This would provide readers with a clearer picture of the trade-offs involved and support claims regarding the method’s efficiency.

**Questions:**

as mentioned above

---

### Meta-Review · Area_Chair_XGrU · 2024-12-20

**Metareview:**

This paper presents a novel gradient-based method for hyperparameter optimization within bilevel optimization frameworks. It aims to address the challenges of existing methods, particularly regarding memory requirements and convergence dependencies. The idea is relevant and could potentially streamline hyperparameter optimization in machine learning applications, contributing to the field.

The reviews received were varied, with one reviewer rating the paper as marginally below acceptance threshold, while the others expressed stronger concerns, leading to a unanimous sentiment leaning towards rejection. Key weaknesses highlighted include the computational efficiency, the lack of detailed runtime analysis, and an insufficient secondary assessment comparing the proposed method to existing approaches, particularly newer datasets and methods.

Despite the authors' rebuttal attempting to address these concerns, significant issues remained unresolved. Reviewers noted that the proposed method still exhibits scalability limitations and may not be practical for larger tasks, which could hinder its applicability. Moreover, the rebuttals did not sufficiently clarify the approximation errors or commit to substantial improvements in the methodology.
In conclusion, the reviewers collectively expressed reservations about the paper's contributions and practicality, leading to a final recommendation for rejection. The concerns outlined, including the lack of comprehensive runtime data and clarity in contributions, have not been adequately addressed in the authors' responses. Given this situation, the final decision is to reject the submission.

**Additional Comments On Reviewer Discussion:**

The reviews received were varied, with one reviewer rating the paper as marginally below acceptance threshold, while the others expressed stronger concerns, leading to a unanimous sentiment leaning towards rejection. Key weaknesses highlighted include the computational efficiency, the lack of detailed runtime analysis, and an insufficient secondary assessment comparing the proposed method to existing approaches, particularly newer datasets and methods.

Despite the authors' rebuttal attempting to address these concerns, significant issues remained unresolved. Reviewers noted that the proposed method still exhibits scalability limitations and may not be practical for larger tasks, which could hinder its applicability.

---

### Decision · Program_Chairs · 2025-01-22

Reject